Chicago sky blue gel for better visualization of Demodex in patients with Demodex blepharitis

Udomwech Lunla 1
Tawanwongsri Weeratian weeratian.ta@gmail.com 2
Mordmuang Auemphon 3
1 Department of Ophthalmology, School of Medicine, Walailak University , Thasala , Nakhon Si Thammarat , Thailand
2 Division of Dermatology, Department of Internal Medicine, School of Medicine, Walailak University , Thasala , Nakhon Si Thammarat , Thailand
3 Department of Microbiology, School of Medicine, Walailak University , Thasala , Nakhon Si Thammarat , Thailand
Guan Fanglin
Electronic publication date: 2023 Nov 17
Publication date: 2023
Volume: 11
Electronic Location ID: e16378
Received 2023 May 17; Accepted 2023 Oct 9
Copyright: ©2023 Udomwech et al.
Copyright year: 2023
Copyright holder: Udomwech et al.
License: This is an open access article distributed under the terms of the Creative Commons Attribution License, which permits unrestricted use, distribution, reproduction and adaptation in any medium and for any purpose provided that it is properly attributed. For attribution, the original author(s), title, publication source (PeerJ) and either DOI or URL of the article must be cited.
License URL: https://creativecommons.org/licenses/by/4.0/

Keywords: Chicago sky blue, Gel, Demodex, Blepharitis

Funding: The authors received no funding for this work.

==============================
Background

Demodex blepharitis is a common chronic disease. The number of mites is associated with ocular discomfort. The accurate number derived from well-stained specimens is, hence, in favor of diagnosing, monitoring, and determining treatment responses.

Methods

A cross-sectional study was conducted between April and July 2022 at the dermatology and ophthalmology clinic, Walailak University, Thailand. Adult participants with clinical suspicion of Demodex blepharitis were recruited. We examined eyelashes under light microscopy to quantify the number of Demodex mites before and after adding CSB gel. The mite counts, evaluated by an untrained investigator and an experienced investigator, were recorded and compared.

Results

A total of 30 participants were included for final analysis, among which 25 (83.3%) were female. The median age was 64.0 years (IQR, 61.0–68.0). The median Demodex counts evaluated by the experienced investigator before and after adding CSB gel were 1.0 (IQR, 0.0–1.0) and 2.5 (IQR, 2.0–3.0), respectively (p < 0.001). Moreover, the median Demodex counts evaluated by the untrained investigator before and after adding CSB gel were 1.0 (IQR, 0.0–1.0) and 2.0 (IQR, 1.0–3.0), respectively (p < 0.001). The correlation coefficient between Demodex counts after the addition of CSB counted by the experienced investigator and those counted by the untrained investigator was 0.92 (p < 0.001). CSB gel is a promising product to identify and quantify the number of Demodex mites. The findings supported the consideration of CSB gel as one of the diagnostic stains.

Introduction

Blepharitis is a chronic inflammatory condition of eyelid margins, commonly found in outpatient clinics (Din & Patel, 2012; Lemp & Nichols, 2009; Rim et al., 2017). It is categorized into two categories: anterior and posterior blepharitis. The etiology of blepharitis is multifactorial, including infectious and inflammatory causes. Common causative pathogens are bacteria and parasites (Din & Patel, 2012). The reported prevalence of ocular Demodex varies widely, ranging from 16–70% (Zhang et al., 2020). In Thailand, ocular Demodex was reported in 42% of the population (Kasetsuwan et al., 2017). Risk factors of Demodex blepharitis include rosacea, diabetes, increasing age, local or systemic immunosuppression, stress, alcohol intake, sun exposure, and smoking (Rhee et al., 2023). Among the patients with detected Demodex, approximately 70–80% had ocular symptoms or signs (Bhandari & Reddy, 2014; Kasetsuwan et al., 2017; Murphy, O’Dwyer & Lloyd-McKernan, 2019). The main symptoms of Demodex blepharitis include ocular dryness, severe itching, burning, foreign body sensation, crusting, redness of the lid margin, and blurry vision (Hom, Mastrota & Schachter, 2013; Luo et al., 2017).

To diagnose, Liu, Sheha & Tseng (2010) proposed diagnostic criteria as follows: (1) clinical features including anterior blepharitis, meibomian gland dysfunction, dry eyes, and/or ocular allergy; (2) slit-lamp examination showing typical cylindrical dandruff at the root of eyelashes; and (3) microscopic confirmation revealing Demodex eggs, larvae, and adult mites in epilated lashes. Furthermore, other related conditions caused by this parasite include conjunctivitis, keratitis, chalazion, madarosis, and trichiasis (Fromstein et al., 2018). Demodex mites were detected and quantified using biomicroscopy, either with or without cylindrical dandruff removal and lateral eyelash traction as non-invasive methods. However, the technique revealed a high false-negative rate (16.9%) and underestimated the number of mites due to some residing deep within the follicle, as well as eggs, larvae, and mites embedded in the removed collarette (Muntz et al., 2020; Tanriverdi et al., 2020). In cases with a high suspicion of serious Demodex infestation, epilation and microscopic examination were therefore inevitable to detect and accurately quantify the exact number of parasites. Microscopic examination of Demodex mite is challenging because of its small size (approximately 200 µm), transparency, residing within the pilosebaceous units, and an artifact of sampling techniques (Nashat et al., 2018). Without a special mount preparation, the negative detection and detection with a lower number of mites were reported in approximately 5–20% and 60% of the patients with blepharitis, respectively (Arici et al., 2021; Kiuchi, 2018).

Several mounting methods and stains have been utilized for better visualization of Demodex, including 10% potassium hydroxide, 100% alcohol, fluorescein dye, and methylene blue (Arici et al., 2021; Kheirkhah et al., 2007; Kiuchi, 2018). Kiuchi (2018) demonstrated that the methylene blue staining method helps detect the presence and quantify the number of Demodex mites on its blue contrast background. However, the process of staining and examining under light microscopy remains challenging in real-life practice since eyelashes are, de facto, lightweight and easily blown away causing loss of specimen. To solve this problem, we thus synthesized the new staining product comprised of Chicago sky blue (CSB) stain in a gel mixture. Using one drop of CSB gel preparation on glass slide, the clinicians can easily place the eyelash in the gel and continue epilating other lashes with no risk of specimen loss and saves time due to the technique’s ease of specimen placement, also allowing convenient transportation to the microscope without any risk of losing the specimen—it proves valuable when the clinical examination room is distant from the laboratory examination room. This water-soluble stain previously proved its usefulness in fungal detection. Having the bluish-to-purplish background, the pathogens were effortlessly identified with a more accurate measure of quantity (Baddireddy & Poojary, 2019; Lim & Lim, 2011; Liu et al., 2015; Prakash et al., 2016).

To our knowledge, there is still a lack of studies to investigate the effectiveness of CSB stain–particularly in the gel preparation–as a diagnostic method of Demodex blepharitis. We hence aimed to assess the effectiveness of in-house CSB gel in identifying Demodex mites under light microscopy in patients with blepharitis. Because the severity of symptoms has been linked to the intensity of Demodex infestation (Cheng et al., 2019; Lee et al., 2010), the management of demodicosis focuses on reducing the mite count to restore the balance of the ocular surface ecology (Martínez-Pulgarín, Ávila & Rodríguez-Morales, 2021). Accurate identification of mite counts using CSB gel could aid in the diagnosis process and monitoring the success of the given treatment.

Materials & Methods

Patients

This cross-sectional study design was adopted between April and July 2022 at the dermatology and ophthalmology clinics at Walailak University, Thailand. During the recruitment process, we posted announcements about this study in the dermatology and ophthalmology waiting rooms and asked for volunteers to participate. To minimize the undue influence, a co-investigator organized the registration and withdrawal processes. Participants with clinical suspicion of Demodex blepharitis were recruited and screened by medical history and physical examination. The inclusion criteria included: (1) being 18 years or older; (2) having suspicious presentations including anterior blepharitis, meibomian gland dysfunction, dry eyes, and/or ocular allergy; (3) being able to read, write, and understand Thai and having the capacity to consent. The exclusion criteria included being unwilling to participate in the study. The written informed consents were obtained from all participants involved in the study. Baseline characteristics were collected including age, gender, comorbidities, and clinical diagnosis. We utilized a slit lamp biomicroscopy (AT 900, Haag-Streit, Switzerland) to zoom in on areas of lesions; and photographs were obtained.

CSB gel preparation

Chicago Sky Blue 6B (CSB) is a new contrast stain that introduces to highlight the fungal hyphae and spores, blue against a purplish background (Lodha & Poojary, 2015; PubChem, 2022a). The 2% CSB gel was prepared separately by a 2-phase formulation. Water phase: 2 mg of CSB powder was dissolved in 20 mL of distilled water. Gel phase: 1 g of sodium polyacrylate (C3H3NaO2) was thoroughly mixed in 50 mL of distilled water until forming the gel solution. The two phases were put together followed by adding water to obtain the volume of 100 mL and homogenizing for 10 min at room temperature. CSB gel was registered in Thailand’s petty patent (No. 2203001836).

Sample collection and staining procedures

After all participants completed a medical history interview and underwent a full clinical examination. The random epilation of four nonadjacent lashes per eye was performed as the standard sampling (Coston, 1967) using flat stainless-steel tweezers (E2160, Storz, Tuttlingen, Germany). After they were placed on a glass slide without adding any solutions, the number of Demodex was counted conventionally under the microscope. One drop of CSB gel was then added to the slide. The slide was covered gently with a cover slip, and the excess stain was removed using blotting paper. The samples were left for 15 min at room temperature and then re-examined under light microscopy to quantify the number of Demodex mites. Microscopic examination was done by one dermatologist and one untrained observer under low power (×100 magnification), and high power (×400 magnification) objectives of a light microscope (Olympus CX33, Japan). The step-by-step specimen preparation is illustrated in Fig. 1. Two separate blinded investigators–including investigator A (a dermatologist) and investigator B (a third-year medical student studying in the preclinical years)–counted the number of mites. Investigator A was selected based on his five years of experience in clinical diagnosis, investigation, and management of patients with Demodex-associated inflammatory skin conditions. The time provided for evaluating was one minute per glass side. All reported numbers of Demodex mites were recorded for analysis. Examples of microscopic photographs of Demodex mites before (A) and after the addition of the CSB gel (B) at 400× magnification are shown in Figs. 2, 3 and 4.

Figure 1 Specimen preparation.

A step-by-step preparation for staining a specimen with Chicago sky blue gel.

Figure 2 Better visualization of Demodex after staining with Chicago sky blue gel.

Microscopic photographs of Demodex mites (black arrows) before (n = 0, 2A) and after (n = 3, 2B) the addition of the Chicago sky blue gel (400× magnification).

Figure 3 Better visualization of Demodex after staining with Chicago sky blue gel.

Microscopic photographs of Demodex mites (black arrows) before (n = 1, 3A) and after (n = 2, 3B) the addition of the Chicago sky blue gel (400× magnification).

Figure 4 Better visualization of Demodex after staining with Chicago sky blue gel.

Microscopic photographs of Demodex mites (black arrows) before (n = 2, 4A) and after (n = 3, 4B) the addition of the Chicago sky blue gel (400× magnification).

Sample size and statistical analysis

The required sample size was calculated for the paired samples t-test based on a previous study (Kiuchi, 2018), with hypothesized mean difference and standard deviation of difference at 1.5 and 2.5, respectively. Using an alpha level of 0.05 and a desired power of 0.80, we estimated that a total sample size of 30 participants, including a 25% loss-to-follow-up rate (incomplete clinical data), would be required to detect a minimum effect size of 0.60 for the primary outcome measure. This study was approved by the Walailak Ethics Committee (WUEC-22-100-01). The ethics committee took into account and complied with the laws of Thailand, including the Personal Data Protection Act (PDPA). In addition, this study complied with the principles of the Declaration of Helsinki and International Conference on Harmonization of Good Clinical Practice. The study was registered in the Thai Clinical Trials Registry (TCTR20220401002). Data files and personal information were encrypted, password-protected, and saved to a secured computer that was only accessible to the study coordinators to ensure confidentiality. The participants could access their own data by directly contacting the study coordinators. No information that could link an individual to the data was revealed.

For continuous data, the mean and standard deviation (SD) or median and interquartile range (IQR) were reported based on the data distribution. Frequencies and percentages were used for categorical data. For inferential statistics, differences in the number of Demodex mites before and after the addition of CSB gel were evaluated using paired t-tests or Wilcoxon’s matched-pairs signed-rank test depending on the distribution of the data. Pearson’s correlation coefficients or Spearman’s rank correlation coefficients were used to assess the relationships between the number of mites counted by investigator A and those counted by investigator B. A p-value of <0.05 in the two-tailed tests was considered statistically significant. Statistical analysis was performed using SPSS software version 18 (SPSS Inc., Chicago, IL, USA).

Results

A total of 30 participants were included in the study, among which 25 (83.3%) were female. The median age was 64.0 years (IQR, 52.0–68.0) and the mean body mass index was 23.3 (SD, 2.8). A majority (73.3%) reported their comorbidities including dyslipidemia (50.0%), essential hypertension (36.7%), type II diabetes mellitus (6.7%), and cerebrovascular disease (10.0%).

The positive rate of Demodex mites counted by investigator A before and after adding CSB gel was 56.7% and 100.0%, respectively. And the positive rate of Demodex mites counted by investigator B before and after adding CSB gel was 66.7% and 88.3%, respectively. Bland-Altman analysis of Demodex counts (shown in Figs. S1–S2) was performed using light microscopy, with and without CSB gel, by investigator A and investigator B to demonstrate the agreement between these two different assays with concordance rates of 90.0% and 93.3% using 1.96SD (95% confidence interval), respectively. The median Demodex count–evaluated by the investigator A–before the addition of CSB gel was 1.0 (IQR, 0.0–1.0; min-max, 0.0–4.0), and was 2.5 (IQR, 2.0–3.0; min-max, 0.0–7.0) after CSB staining (p < 0.001). The median Demodex count–evaluated by investigator B–before the addition of CSB gel was 1.0 (IQR, 0.0–1.0; min-max, 0.0–2.0), and was 2.0 (IQR, 1.0–3.0; min-max, 0.0–6.0) after CSB staining (p < 0.001). Bar plots demonstrating the means of Demodex mites counted by both investigators were shown in Figs. S3–S4. The correlation coefficient between Demodex counts before the addition of CSB counted by the two investigators was 0.51 (p = 0.004). Furthermore, the correlation coefficient across Demodex counts after the addition of CSB gel counted by both investigators was 0.92 (p < 0.001).

Discussion

This cross-sectional study aimed to assess the effectiveness of in-house CSB gel in identifying Demodex mites under light microscopy in patients with blepharitis. Its effectiveness in the naïve and expert investigators was compared. We found that CSB gel significantly enhances the visualization of Demodex mites both in the naïve and expert investigators with a very strong correlation of mite counts.

The two Demodex species, D. folliculorum and D. brevis, are commonly present on the normal skin of adult humans, particularly in pilosebaceous units on face and neck (Aktaş Karabay & Aksu Çerman, 2020). D. folliculorum is mainly clustered around the eyelash root and follicles in the eye, while D. brevis is solitary and resides in meibomian and sebaceous glands near the eyelash follicles (Ye et al., 2022). Mature individuals of D. folliculorum typically reach a length of about 0.3–0.4 mm, while those of D. brevis typically measure around 0.2–0.3 mm (Bitton & Aumond, 2021). The length of male D. folliculorum may be similar to that of female D. brevis (Desch & Nutting, 1972). Two distinct species can be distinguished by examining their gnathosoma and podosoma. Despite both possessing four pairs of podosoma located in the anterior third of the body, only D. folliculorum exhibits spurs on each leg. Furthermore, D. folliculorum features a longer posterior end and a complex gnathosoma with a supracoxal spine, while D. brevis displays a shorter posterior end and a simpler gnathosoma. In terms of mouthparts, D. folliculorum presents a complex array of structures, including a round oral opening, a sharp oral needle, and a unique hypostome resembling a longitudinal spindle positioned centrally, with seven robust palpal claws on each side of the mouthparts. Conversely, D. brevis lacks an oral needle and possesses only five pairs of palpal claws on the terminal segment of the palpus (Jing, Shuling & Ying, 2005; Kosik-Bogacka et al., 2013). These mites lack a terminal excretory opening, and undigested material is regurgitated, mixing with epithelial cells, keratin, and eggs to form cylindrical dandruff. These deposits contain proteases and lipases, leading to symptoms of irritation. The mites and their debris trigger inflammatory cascades (Fromstein et al., 2018). Additionally, these mites possess their own microflora, including Streptococcus spp., Staphylococcus spp., and Bacillus Oleronius. The loading of bacterial antigens results in the stimulation of host inflammatory responses (Nicholls et al., 2017). In asymptomatic healthy individuals, the prevalence of Demodex mites ranged from 16.2–43.5%, while in patients with blepharitis, the prevalence of Demodex mites ranged from 62.4–89.3% compared to those with a healthy status (Akkucuk et al., 2023; Bitton & Aumond, 2021; Correa Fontt et al., 2020; Kasetsuwan et al., 2017).

A diagnosis of facial demodicosis is made when of more than five adults, larvae or eggs per cm2 are found with a skin-surface biopsy (Forton & Seys, 1993). In papulopustular lesions, demodicosis is diagnosed when there is a density of Demodex mites, at three or more mites per five pustules (Huang, Hsu & Lee, 2020). However, the gold-standard diagnostic method or cut-off density of ocular infestation has not been well delineated (Zhang et al., 2020). Demodex infestation is more prevalent in the elderly (Sędzikowska, Osęka & Grytner-Zięcina, 2016; Wesolowska et al., 2014) and in individuals with blepharitis (Biernat et al., 2018). The number of Demodex showed significant positive correlations with ocular discomfort (Lee et al., 2010). Thus, the special preparation of specimens with a high accuracy for detection of Demodex quantities might play a crucial role in diagnosis and follow up of patients with Demodex blepharitis. Potassium hydroxide and alcohol were added to a specimen aiming for better visualization (Arici et al., 2021). However, the transparency of Demodex mites and residing in hair follicles in a cluster consisting of several mites make the quantification of mites challenging. Even fluorescein stain might not provide satisfactory contrast between mites and the background (Kheirkhah et al., 2007). Kiuchi (2018) demonstrated that the Demodex mites were easily detected on the blue background of the methylene blue dye. Yet, eyelashes are lightweight and non-cohesive to a glass slide. Our in-house CSB gel, therefore, became a promising agent for clinical diagnosis and point-of-care applications. CSB is an ionic azo compound classified in the chemical class of benzamide dyes. Its molecular weight is 992.8, and its molecular formula is C34H24N6Na4O16S4. It is a water-soluble dye that is bright blue in color, with a concentration of up to 40 mg/mL (PubChem, 2022b). Unlike methylene blue, CSB has not been reported to cause delayed cutaneous effects or photo-irritant contact dermatitis (PubChem, 2022a; PubChem, 2022b).

We acknowledge several limitations. First, the efficacy of Demodex visualization using other stains, including potassium hydroxide, Löffler’s alkaline methylene blue, and fluorescein dye, was not compared with CSB gel in our study. Further studies are needed to assess and compare the effectiveness of these useful stains. Second, the entire follicular unit was not removed during epilation, and the Demodex mites remaining in those epilated hair follicles were not counted. Therefore, using this method could result in an underestimation of the number of mites. Third, there is still ongoing controversy regarding its contribution to the development of ocular disease. The majority of studies conducted were cross-sectional in design (Zhao et al., 2012). Further studies with a prospective clinical design, both before and after treatment, should be done to identify it as the causative organism. Fourth, the assessment of changes in the mites’ morphology and size after the addition of CSB gel was not conducted. Further studies are necessary to assess these changes over a specific time period after the addition of CSB gel.

Conclusions

CSB gel is a promising product to identify and quantify the number of Demodex mites. Its cohesive property eases specimens handling and visualization under light microscopy. The findings supported the consideration of CSB gel as one of the diagnostic stains.

Supplemental Information

Data S1 Raw data

Click here for additional data file.

Figure S1 Bland-Altman analysis of Demodex mites was performed using light microscopy, with and without Chicago sky blue gel, by investigator A to show the agreement between these two different methods

Each dot represents one data point. The concordance between these diagnostic tests was examined using 1.96SD (95% CI) represented by solid line.

Click here for additional data file.

Figure S2 Bland-Altman analysis of Demodex mites was performed using light microscopy, with and without Chicago sky blue gel, by investigator B to show the agreement between these two different methods

Each dot represents one data point. The concordance between these diagnostic tests was examined using 1.96SD (95% CI) represented by solid line.

Click here for additional data file.

Figure S3 A bar plot showing the means of Demodex mites counted by investigator A

The mean number of mites using light microscopy with and without Chicago sky blue gel was 0.93 (SD 1.08) and 2.87 (SD 1.63), respectively.

Click here for additional data file.

Figure S4 A bar plot showing the means of Demodex mites counted by investigator B

The mean number of mites using light microscopy with and without Chicago sky blue gel was 0.80 (SD 0.66) and 1.47 (SD 1.04), respectively.

Click here for additional data file.

We are immensely grateful to Mr. Danuwat Pholsin for assistance with data evaluation.

Additional Information and Declarations

Competing Interests

Author Contributions

Human Ethics

Patent Disclosures

Data Availability

The authors declare there are no competing interests.

Lunla Udomwech performed the experiments, prepared figures and/or tables, and approved the final draft.

Weeratian Tawanwongsri conceived and designed the experiments, analyzed the data, prepared figures and/or tables, authored or reviewed drafts of the article, and approved the final draft.

Auemphon Mordmuang analyzed the data, authored or reviewed drafts of the article, and approved the final draft.

The following information was supplied relating to ethical approvals (i.e., approving body and any reference numbers):

This prospective study was approved by the Walailak Ethics Committee (WUEC-22-100-01).

The following patent dependencies were disclosed by the authors:

CSB gel was registered in Thailand’s pity patent (No. 2203001836).

The following information was supplied regarding data availability:

The raw measurements are available in the Supplementary Files.

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
