# Peer review of "Chicago sky blue gel for better visualization of Demodex in patients with Demodex blepharitis"

_PeerJ, doi:10.7717/peerj.16378_

## Round 0.1 · original submission · Major Revisions

The three reviewers all made suggestions to your manuscript to varying degrees, including suggestions for further improvement on the part of the experimental design. I hope you will take it seriously and reply to these comments one by one.

·

Basic reporting

a. English language used is clear, unambiguous, and easily understandable.
b. Introduction: The flow of information is not appropriate like it should begin with the rate of infection or occurrence, its symptoms, presently available diagnostic methods and treatment if required. But here its mentioned occurrence first followed by some diagnostic methods then symptoms and treatment. An article by Muntz A et al 2019 mentions about non invasive method of diagnosis of demodex which is not mentioned in this article. Fluorescein stain and biomicroscopic methods (Tanriverdi C et al 2020) are also used for demodex diagnosis which is not mentioned in this article.
c. Chicago sky blue gel: The necessity to use CSB as is not mentioned. Most of the CSB articles have used CSB solution with KOH to get contrast. So can you please mention the advantage of using CSB as gel instead of solution.

Experimental design

a. Why invasive method of diagnosis was chosen when non-invasive method was available.
b. Why only CSB was used and why Löffler’s alkaline methylene blue or fluorescein were used to study the difference or significance.
c. Sample size calculation: In line no 127 it is quoted that “Based on a previous study (Kiuchi 2018), with an alpha of 0.05 and power of 80%.” Can you please explain this how did you get 30 as sample size.
d. Methodology: The number of eyelashes removed to do light microscopy is not mentioned. The solution used for light microscopy is not mentioned in line no 112 and 113.

Validity of the findings

a. Results shows correlation between experienced vs unexperienced person. The IQR of demodex identified in light microscopy and CSB is almost the same. Whereas when we check the total number of demodex identified in light microscopy and CSB, there is a big difference between the experienced(n=28 & n=86) and unexperienced (n= n=24 & n=44). In what way is the CSB much better for unexperienced persons.

Additional comments

a. Confidentiality, conflict statement and patency has been mentioned.
b. Conclusion says findings encourage the use of CSB gel as a diagnostic method but I would consider the authors to specify that CSB can be considered as one of the stain as a diagnostic method.
c. Limitations: the bias in sample collection is not mentioned as some demodex is retained in the pores of eyelashes. So they will not be counted. Only one stain was used so superiority of the stain cannot be commented.

Reviewer 2 ·

Basic reporting

Demodecid mites are common in humans, often asymptomatic, but sometimes can cause various types of disease symptoms and chronic ailments. Detection and identification are complicated, due to the hidden lifestyle and microscopic size (reduction of morphological elements). However, not only the detection of these mites, but the determination of their number (density) is an important diagnostic criterion. Hence, the introduction of new methods to facilitate diagnosis is needed.

The conducted research is interesting and has practical significance. The manuscript requires some corrections (improvements) and clarifications.

Title and Abstract - Demodex should be in italics

Lines 44-48 Did the cited review of studies concern the mere occurrence (prevalence of infestation) of Demodex spp. in the eye area? Has the relationship of its presence with blepharitis been demonstrated? The mere presence of these mites is not proof of disease; demodecid mites may occur asymptomatic, even in all subjects from a given population (detection - the question of the accuracy of the test procedure); the development of disease symptoms is usually related to their greater number (density) and individual immunity of the person / patient.

Lines 169-172
Data on the occurrence (location, topography) of two Demodex species in humans are imprecise and require updating; data on typical location should be based on asymptomatic (typical for these mites) occurrence - there are more recent data and publications discussing more precisely the differences in the occurrence of these species.

Lines 172-78
“ The body is covered with scales for anchoring itself in the hair follicle, and the mite has pin-like mouth parts for consuming epithelial cells…” Strange, untrue and unscientific description - you should rather use specialist (acarological) literature on these mites.

Experimental design

The manuscript presents the possibility of using a new, simple and effective method useful in the detection of demodecis mites, which may be the cause of blepharitis. The method is not original, it has been used before to detect other pathogens, but new possibilities of its use have been shown; it seems that it may also have wider applications, also in detecting other types of demodicosis.
The methodology used in the work is simple, transparent, the study seems well planned; the selection of the researchers conducting the observations (including the researcher described as an experienced dermatologist - lines 116-122) seems somewhat unclear (subjective) - whether it was a person experienced in detecting and identifying demodecid mites (what criterion was used to assess the researcher's experience)?

Validity of the findings

The study concerned the verification of the usefulness of the diagnostic method for the detection of demodecis mites, which may be valuable in ophthalmological diagnostics.
However, it would also be important to specify the Demodex species; in humans there are 2 species - Demodex folliculorum and D. brevis, with different topical preferences (they live in different skin structures). However, they can coexist in the same areas of the skin (including the eye area). However, D. brevis is smaller and less frequently detected. Studies on the relationship between demodecid mites and blepharitis often lack information on which species was the etiological factor here. Different species of the family Demodecidae may cause different symptoms and, moreover, may require a different therapeutic approach (different control methods), hence species identification should be the basis for diagnosis.
Species identification would therefore be valuable here, also in the context of knowledge about the correlation of blepharitis with the occurrence of individual species of demodecid mites.
And here the question arises - can the use of a dye that facilitates the detection (and counting of specimens) of these mites facilitate or hinder the observation of species characteristics (i.e. species identification)?

Additional comments
* * *
Reviewer 3 ·

Basic reporting

Dear authors,
I have found your article very useful for clinical diagnosis. However, I would recommed to include two additional analyses to compare the outcomes of Demodex detection.

Experimental design

If you want to compare between methods (with and without gel) you have to explore the agreement through a Bland Altman regression. See an example

Aleuy et al. 2023.Tissue-specific assessment of oxidative status: Wild boar as a case study.Frontiers in Veterinary Science. 10. 10.3389/fvets.2023.1089922 .

Once both methods are interchangeable, you may explore whether counts based on CSB gel provided more Demodex count (i.e., parasite load), that in turn is a proxy for disease severity.
This analysis could be done using a simple count comparison and a Mann-Whitney U test comparing the counts with and without gel. A bar plot showing mean counts between techniques would be also recommended.

Validity of the findings

Findings are valid

Additional comments

None

---

## Round 0.2 · Minor Revisions

Although your responses received positive ratings from most reviewers, then one reviewer raised new questions and comments about the reliability of your study. Based on this, I suggest that you need to fully consider these suggestions, and make targeted replies and revisions.

·

Basic reporting

No comment

Experimental design

No comment

Validity of the findings

No comment

Additional comments

No comment

Reviewer 2 ·

Basic reporting

(Also commentary to Validity of the findings and Experimental design)

All my comments and suggestions have been taken into account. However, in one issue (in my opinion, key in the context of research reliability), the answer and additions are not satisfactory to me, they are rather a shortcoming of the research.

The review included a question about the importance of the diagnostic method used in the context of Demodex spp. species identification - the question was whether it could show features important for species identification or otherwise (features become less visible or deformations may occur).
In laboratory diagnostics (medical, veterinary), species identification is often carried out incorrectly, which, unfortunately, the authors themselves refer to in the Discussion - "The posterior two-thirds part, called the opisthosoma, is used to distinguish the species...". Well, this is not true (or rather has historical significance) - according to the current state of knowledge, in the light of research on the biodiversity of Demodecidae (discoveries of new species in various hosts, e.g. in the cat, 4 co-occurring species are currently known, similarly in the dog; no greater species diversity in humans can be excluded), as well as taxonomic analyzes (including the variability of metric features and body proportions), neither sizes nor proportions can be used for species identification (and if they are - only as a supplement, e.g. when examining a large sample, after an initial identification of a representative sample of specimens based on appropriate taxonomic characteristics). Reliable features for species identification are primarily structures located on the gnathosoma and podosoma. Proportions and dimensions (e.g. length of opisthosoma) are highly variable; in addition, in the case of human demodecids, the fact that male D. follicurorum may have opisthosoma proportions similar to female D. brevis is often not taken into account.
Various substances used in diagnostics can cause changes in size (shrinkage, deformation) of soft parts of the demodecid mites body, especially opisthosoma. Hence the question - did the authors experimentally check whether the administration of CSB gel can cause any changes in shape and size? Hence the question in the review, what, according to the authors, meant an experienced researcher in terms of identifying Demodex. Although a dermatologist will undoubtedly identify the Demodex mite (or at least the Demodecidae family), only a few have the skills and experience to correctly identify the Demodex species (this is rather the scope of the acarologist's specialty). Therefore, in many scientific publications, the authors limit the identification only to the genus Demodex (without distinguishing between species) and this is real/reliable. Species determination is of course important as different specific species may cause different symptoms in the host; furthermore, an infestation with an atypical species cannot be ruled out. However, in diagnostic practice, identification at the species level is not always important, because it rarely translates into therapeutic management. It is different in scientific research (for publication in a scientific journal) - here identification should be reliable, based on standards according to the current state of science, and not better or worse common practices.
Hence, if there was no species identification confirmed by an actual specialist (not only based on unreliable metric features), nor verification of how the method used affects Demodex specimens (e.g. whether it does not cause opisthosoma shrinkage), the authors should limit the data only to information about genus Demodex, without information on individual species.

Experimental design

ad Basic reporting

Validity of the findings

ad Basic reporting

Additional comments
* * *
Reviewer 3 ·

Basic reporting

I have no additional comments

Experimental design

--

Validity of the findings

--

Additional comments

I have no additional comments and the ms could be accepted for publication

---

## Round 0.3 · accepted · Accept

We greatly appreciate your understanding and cooperation throughout the revision period. The revisions you made based on the reviewers' comments have significantly improved the quality and clarity of your research. As well, we would like to extend our appreciation for your dedication and effort in preparing this manuscript. We look forward to future collaborations and hope to have the chance to publish more of your research in our journal.